# The Determining Factors of the Delocalization of Italian Companies to East European Countries

**Klodian Muço and Emiljan Karma \***

Research Centre on Economics of Transition Countries, Catholic University Our Lady of Good Counsel, 1000 Tiranë, Albania; k.muco@unizkm.al
* Correspondence: e.karma@unizkm.al

**Abstract:** Reflections on the factors that influence the delocalization of Italian companies to other countries, arising from a rapidly changing world economy, offer invaluable insight into companies facing similar tough choices. Usually, this decision is attributed to economic, fiscal, and institutional factors. This article examines the process of delocalization of Italian enterprises and empirically verifies the main factors which influence the delocalization of Italian enterprises to East European Countries. The results suggest that "labor market regulation", "business regulation", and "size of government" or better said, the labor cost the host country, is the main incentive of Italian enterprises to delocalize their production. Empirical results also show that institutional factors such as "rule of law", "control of corruption", "political stability", and "broadband infrastructure" have a positive and significant correlation with the delocalization of Italian enterprises to the East European Countries. The results in this case suggest that Italian companies are looking for an environment in which the state knows how to enforce their own rules, but also a state that is somewhat "corruptible".

**Keywords:** delocalization; organization of production; East European Countries; Italian companies; offshoring; labor market regulation; instructional factors

## 1. Introduction

One of the phenomena that have affected all the main industrialized countries in the last two decades has been the international division of labor (Berger 2006; Hanson et al. 2003) characterized by enterprises that separate the phases of their activities into different modules and delocalize these modules into different countries in order to take advantage of the different location conditions, of the differences in production costs, and especially of the labor costs.

A European Restructuring Monitor Report (2007) report shows that delocalization of production, and above all that of the manufacturing sector, is an ever-growing phenomenon. About 60% of job losses in the manufacturing industry in the EU have resulted from companies that delocalize their production.

In fact, over the last decade, over twenty-seven thousand Italian companies have outsourced production abroad, creating over 1.5 million foreign jobs and leaving to the state 15 billion euros for social safety nets. On a closer overview, only 10% of these companies have gone beyond the European borders (above all in Asia) while the remainder are established in Europe: Austria, Switzerland, Germany, and, in particular, in the Balkan countries, which recently are demonstrating strong growth potential and appear to be sufficiently stable from an institutional point of view. Over 900,000 people work in Italian companies based in the Balkan area, of which 800,000 are in Romania alone (European Restructuring Monitor Report 2007).

According to the OECD (OCSE 2007), there are two main factors that drive delocalization: market access for export companies and low unit cost of labor in neighboring countries.

Given these considerations, we think it is important to understand and analyze the factors that influence the delocalization of Italian companies to the East European Countries.

The main goal of this paper is to understand and to identify the factors affecting the Italian enterprise delocalization and whether there are other factors (not only economic factors) that drive enterprises to delocalize production to a host country.

This study provides firstly a review of empirical studies on delocalization, followed, by the economic factor impact analysis ("labor market organization", "the business regulation", or "size of government") and institutional factors ("control of corruption", "rule of law", "political stability", and "broadband infrastructure") on the delocalization of Italian companies to East European Countries (Albania, Bulgaria, Bosnia and Herzegovina, Croatia, Czech Republic, Estonia, Georgia, Hungary, Latvia, Lithuania, Moldova, Montenegro, Poland, Romania, Serbia, Slovak Republic, and Slovenia).

The methodology used in this empirical analysis is based on panel data at the macroeconomic level, from the official database of the Italian Trade Agency, World Bank, and Economic Freedom of the World, in which the dependent variable is the number of Italian companies in relation to the population. To achieve the goal of this study, panel data regression analysis was considered to define differences between countries. The next step is to explore how the variables considered above (labor market organization, business regulation, size of government, control of corruption, rule of law, political stability, and broadband infrastructure) can influence the choices of Italian investors.

The main findings of this study show that the delocalization of Italian companies to the East European Countries is not only influenced by labor cost, as shown by most of the literature (Helg and Tajoli 2005; Baldone et al. 2001; Cietta 2008; De Nardis and Traù 2005), but also by institutional factors such as the rule of law, corruption, and broadband infrastructure. In relation to this conclusion, it can be stated that the accession of some countries to the European Union has discouraged the delocalization of Italian enterprises to these countries (such as Romania and Bulgaria), as accession causes a reduction in the economic benefits that lead to delocalization.

The study is organized as follows: after the introduction there is a review of the literature on delocalization and a review of some data on the delocalization of Italian companies.

We give a brief description of commercial exchanges between Italy and the East European Countries, in order to highlight trade relations between the two "factor groups" taken into consideration. Finally, the chosen methodology and study indicators are summarized in a synthetic way by analyzing them by means of descriptive statistics and empirical analysis. We finish with the comments on the results of the testing and some final conclusions.

## 2. Literature Review

The process of delocalization has grown significantly in recent years, generating another process that characterizes most of the developed EU countries, particularly Italy: the "de-industrialization" process that leaves Italy increasingly vulnerable to the economic crisis, as well as more "incapable" of offering new generations a job prospect and the possibility of financial wealth.

The opening of the international markets, along with a significant reduction in transport and communication costs, has resulted in a higher integration of economies, accelerating exchanges, investments, and the distribution of production in many markets, and, above all, has driven the economies in question to pursue labor cost advantages in emerging countries.

According to Casaburi et al. (2009), there is a strong link between international fragmentation of production and entrepreneurial success. The companies which choose to delocalize are, according to him, the largest and most productive ones, perhaps because of the need to cope with the fixed costs of the offshoring process.

Delocalization of production is of two types: offshoring of intermediate goods production (vertical) and production of final goods (horizontal).

Usually, the delocalization happening in the Balkan area is one of labor-intensive production processes with intensive labor phases (textiles and clothing). Most of these companies tend to be increasingly involved in vertical offshoring.

According to the OECD (OCSE 2007), there are two main factors that encourage enterprises to delocalize: market access for export companies and lower unit labor cost in neighboring countries. The labor cost is also evidenced in various empirical studies which consider it the main factor which explains the international division of production processes (Helg and Tajoli 2005; Baldone et al. 2001; Cietta 2008; De Nardis and Traù 2005).

Reduction in labor cost according to Giusti (2006) has a positive impact on increasing competition between countries; this leads Italian firms to see the delocalization of production as the only way to remain competitive in the market.

When talking about the cost of labor, in fact, we should take into consideration productivity. According to some OECD estimates, Poland has the highest productivity of any Balkan country and of any former communist country. As for the cost of working hours in euros for companies with more than 10 employees, Albania is the country which has the lowest cost of the working hour. According to the data published by the National Statistical Institutes of the Balkan countries and according to Eurostat data (Eurostat 2016), it appears that the country with the lowest average gross salary of any Balkan country is Albania, with 419 euros, while the country with the highest average gross salary is Slovenia, with 1756 euros.

The overall average gross salary of the 12 countries taken into consideration is 773 euros, whereas the average gross salary in Italy is 2536 euros.

However, the cost of labor is not the only incentive that encourages enterprises to delocalize, there are other factors, such as, distance, flexibility at work (Rodrik 1997), similarity to local culture and traditions, the ability of the investing company to speak the language of the host country, as well as the political and institutional stability of the host country (Helpman 1984; Helpman and Krugman 1985). There are also differences in the fiscal pressures between the EU countries (Rabushka 2003; Mitchell 2004) as well as the pressure from trade unions (Faro 2008; Perulli 2011).

According to a survey of manufacturing companies (sampled subset of 213 from 362 companies that have done offshoring) conducted by Unicredit in 2010 (Amighini et al. 2010), results show that the main factors that encourage manufacturing companies to delocalize are: low labor costs 49.3%, on-site availability of low-cost raw materials 20.7%, proximity to the markets 22.5%, fiscal advantages 17.4%, fewer restrictions on environmental protection and labor law 8%, others 7%.

The role of labor costs as a main factor appears clearly also in the work of (Smith and Pickles 2011; Zhelev and Tzanov 2012). They take into consideration countries before and after they joined the European Union and find that, before joining the EU, Romania and Bulgaria were the most important host countries to which EU companies delocalized production. With their entry into the EU, there was an initial first phase in which production and export to the countries of the EU moved from more expensive countries like Poland to Romania and Bulgaria. Consequently, this led to an increase in salary levels, fiscal burdens, and expectations of further salary pressure in the coming years, which, in turn, led to a slowdown in delocalization to these countries, preferring instead countries that were not part of the EU. This was the first slowdown in delocalization to these countries, although the increase in salary, fiscal, and trade union pressure was partly offset by a significant increase in exports from these countries to other EU countries. Instead, countries like Romania and Bulgaria, by now part of the EU, have begun to specialize in producing superior quality products that require at the same time faster shipping times.

In a similar study, Zhelev and Tzanov (2012) confirm the fact that the entry into the EU of the Balkan countries somehow restrains the further growth and delocalization of EU companies to countries that are about to join the EU.

Landesmann and Wörz (2006) show that, over time, developed countries that are part of the EEC specialize in high-tech industry, whereas the low-tech industry and medium-tech

industry are disappearing from the developed countries which have improved competition in the medium–high-tech industry. At the same time, the less developed countries of the EU such as Bulgaria, Romania, and Croatia show a specialization of exports in the field of basic technologies and in labor-intensive ones.

At the same time, we see a reduction in the deficit in high-tech areas, or rather there is an improvement over time also from this point of view. One reason might be that, over time, the EU states tend to move the production of low-tech products to the less developed countries (Sass and Szanyi 2012). In his study Richet (2013) also confirms this when he shows that delocalization and FDI make an important contribution to integrating economies by creating links between Western multinationals and their regional branches. Over time, according to Richet, in the Eastern countries there is an improvement in the standard of living and a reduction in the overall East–West technological and social gap. This improvement in the standard of living according to Crestanello and Tattara (2006) is also because, when there are investments in a country, linkages are created with other economic activities that are linked to one another (Hirschman 1977).

For Kalogeresis and Labrianidis (2010) sometimes the delocalization brings more development in the host country than in the country of origin. Because it implies a weakening of the ties with the domestic economy, thus, in general, in the case of delocalization it is difficult to identify the "winners" and "losers". Moreover, if a company tries to cut costs to increase productivity and efficiency, in theory, this should also create new jobs. Often, however, the increase in productivity by cutting costs leads only to short-term effects. Salaries and working conditions will hardly hinder the acquisition and retention of skilled labor regarding efficiency and flexibility; and they rarely encourage the company to "invest" in human capital to make it more productive (Blomström and Kokko 1996).

Blomström and Kokko (1996) and (Viesti and Prota 2007) show that the delocalization has a positive impact on the increase in productivity of the multinational companies that delocalize, but at the same time, they also find local subcontractors, which leads to job losses in their country of origin.

Viesti and Prota (2007) and (Labrianidis 2001) in their study conclude that a part of the production is delocalized abroad to Albania and Romania but, at the same time, the productive structure of those small suppliers who, in the past, have had contractual relations with customers of medium and medium-high targets and who today are the most loyal and reliable contractors for the final users, both for delivery times and for sartorial technique, has remained intact. On the contrary, such operators could be the fundamental productive resource to guarantee the re-launch of "Made in Italy" which is already on the horizon of the new emerging markets, especially that of China.

Today, a vicious circle has been created, in which delocalization leads to the closure of third-party service providers and, in turn, the lack of production capacity in the local system leads to greater recurrence of decentralization. This situation has also favored the phenomenon of reliance on Chinese labor located in the territories of Puglia, of strictly "submerged" production segments: the Chinese who work informally in Puglia are able to compete with Albanian and Romanian producers.

Horgos and Tajoli (2015) reach the same conclusions in their study about Germany, which links the effects of offshoring with the skill ratio. The effect depends on the industry, whether it is high-skill production or low-skill production, but nevertheless, there is no evidence for an effect on unemployment.

The loss of jobs due to delocalization does not emerge even in the study of Falzoni and Tajoli (2008) and other international empirical studies (Riess and Uppenberg 2004; Feenstra and Hanson 1996; Amiti and Wei 2004).

With regard to data on companies that have delocalized production abroad, the ICE (Italian Trade Agency) admits that there are no official data on this, only some estimates. Because, according to them, this is a responsibility of the host countries. That said, we can say that, according to some estimates published on its website by the Confindustria Balcani (2021), there are 27,000 companies that have delocalized production abroad and created

more than 1.5 million jobs outside Italy. Of these companies, only 10% have moved beyond the European borders, the rest have moved to other European countries, but especially to the Balkan countries. The CGIA of Mestre (Small Enterprises Artisanal Association) also confirms that the companies that have transferred a part of their production activity abroad number over 27,000. This phenomenon, according to them, has increased by 65% in the period 2001–2011 and, in the same period, the jobs created by these companies over the border number 1,557,000.

Confartigianato reviews the numbers by stating that the workers and employees outside the Italian borders who work in delocalized Italian companies number 835,000 and the companies in question have a total turnover of 217 billion euros. According to them, the manufacturing sector has the highest degree of active internationalization at 22.3%, that is double compared to what was observed for the total Italian economy (10.7%) and triple compared to other sectors (7.3%).

In Europe, on the other hand, the percentage of the delocalization of production in the manufacturing sector is much higher (between 30 and 45%) in small countries with tertiary economies. In the larger European countries, the percentage is lower (between 20 and 25%) and presents different trends among large countries, higher in Germany and Spain and less in Italy and the UK (Amighini et al. 2010).

To conclude, we must also say that, in addition to the offshoring process, there are also companies that have been reshoring, i.e., they have reintroduced manufacturing jobs back to the country of origin. According to the European Restructuring Monitor Report (2007) from 1997 to 31 December 2013, there were 79 companies that had been reshoring. The main reasons for this decision can be summarized as: difference in labor costs is no longer high as before, logistics costs tend to increase more and more, higher requests for "Made in Italy", the distance between the research center in Italy and the production abroad does not allow us to respond quickly to market changes, etc.

## 3. Commercial Exchanges between Italy and the East European Countries

Political and economic relations between Italy and most of the East European Countries represent a strategic priority in Italian foreign policy. This is due to the political tradition, the geographical position, but also due to the cultural affinity.

The strengthening of institutions in the East European Countries and the progressive integration into the EU are important factors for the stability of the European continent.

Referring to the data published by the World Bank, the main East European Countries (Albania, Bosnia and Herzegovina, Bulgaria, Croatia, Macedonia, Montenegro, Romania, Lithuania, Latvia, Moldova, Georgia, and Slovenia) have a relatively low population compared to other European countries (69 million) and a total nominal GDP of 524 billion dollars (IMF Database 2021).

In fact, Italian exports' share to the East European Countries have increased from 7.63% in 2001 to 10.71% in 2008; in monetary value, Italian exports to these countries more than tripled in the period in question and, after a sharp decline of about 25% in 2009, they started to recover again, reaching again 10.08% of the total volume of Italian exports to the Balkan countries (in 2016). However, it must be said that, in terms of monetary value, Italian exports in 2016 are still lower than in 2008 (Figure 1).

Italian imports' share from the Balkan countries rose from 5.73% in 2001 to 7.55% in 2008 and 10.98% in 2016. The highest increase in terms of monetary value was in the period 2001–2008, during the period in question the volume of Italian imports from these countries has more than doubled. The decline in 2009 was slightly higher than the decline in exports (26.5%). The recovery in the following years was higher compared to the exports. If the latter in 2016 are 14.5% less than in 2008, imports in 2016 are 6% higher than those in 2008 (Figure 2).

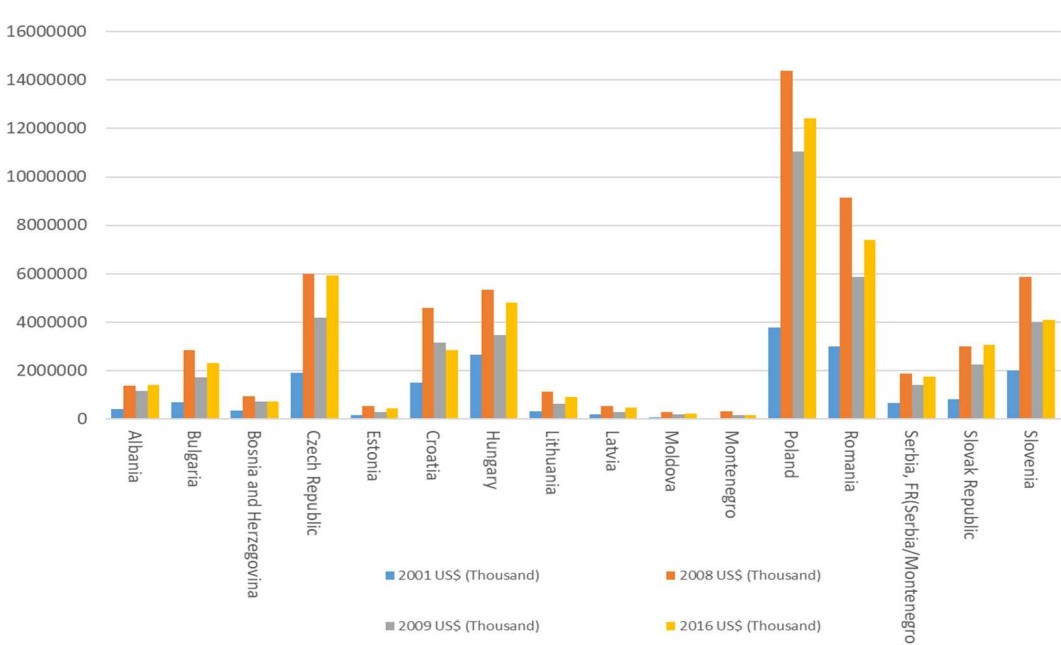

**Figure 1.** Italian Exports to EEC (USD thousand). Source: WITS (2021).

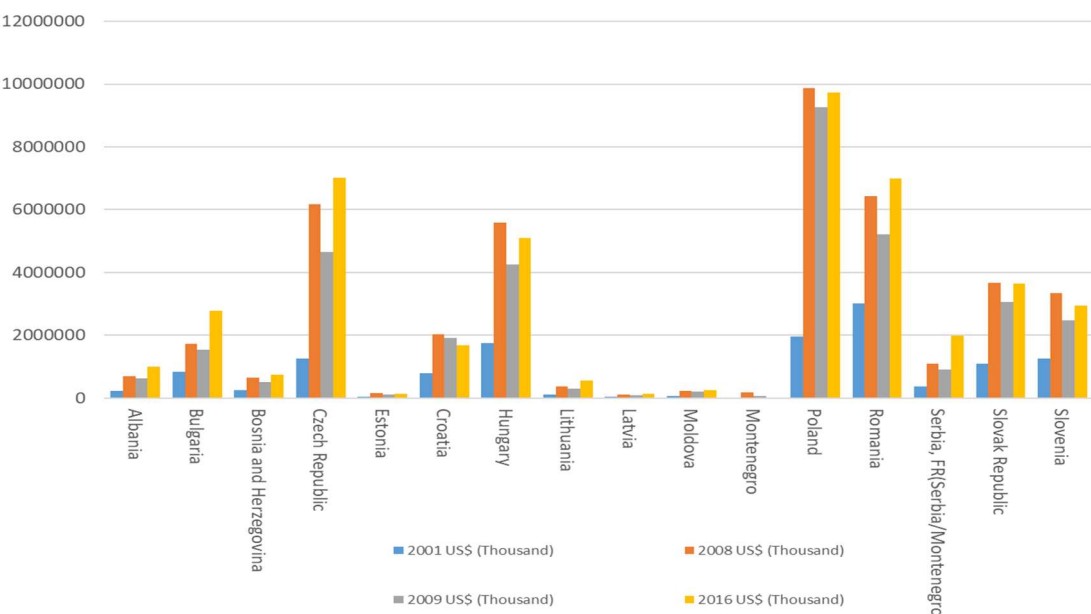

**Figure 2.** Italian Imports from EEC (USD thousand). Source: WITS (2021).

During this period there was a general decline in Italy's export quotas to EU countries: the quotas towards Romania went from 6.22 billion (2008) to 5.81 billion (2012); those towards Bulgaria from 1.93 billion euros (2008) to 1.59 billion euros (2012); and, finally, those towards Croatia from 3.13 billion euros (2008) to 1.98 billion euros (2012). Instead, export quotas to non-EU Balkan countries have progressively increased.

If we refer to the data published by the UN Comtrade Database (2021), in recent years, Italy has established new trade relations with the East European Countries and a sizable portion of its existing trade volume has shifted towards these countries.

Regarding the imports and exports of the individual countries of the East European Countries, for Albania, Italy remains the most important trading partner for both imports and exports. Albanian exports to Italy decreased from 71% of total export volume in 2001 to 62% in 2008, reaching 43% in 2013 to recover again in 2016 to 55%. However, it must be

said that exports to Italy have always been increasing even if, over the years, the weight in percentage has "decreased" in monetary value.

The exports from Croatia and Slovenia went from 24% and 20% in 2001 to 19% and 18% in 2008 and reached 14% and 13% in 2016. Other countries also have had the same trend, i.e., around 10–12% of their exports go to Italy. In recent years, even for these countries, the volume of exports to Italy has decreased in percentage; however, in terms of monetary value, exports have always been increasing, except for 2008.

As for the imports of these countries from Italy, the trend is quite similar to that of exports to Italy. Except for Albania, which imports about 30% of the total volume of imports from Italy, all the other East European Countries have a trend in imports from Italy quite similar to their exports to Italy.

Overall, the import–export of Italy from and to different East European Countries has been increasing before the crisis. At the beginning of the crisis, we see a sharp slowdown in trade in the mentioned countries.

From the analysis of the total flow of trade for these countries, it seems that the crisis for Italy starts in 2008 whereas for most of the East European Countries (especially those outside the EU) the crisis begins one year later. According to Bartlett and Prica (2013) the reason for this is because the economy of the countries in question is strongly linked to that of the countries of the EU, thus the crisis in most of the East European Countries (Balkan countries) has been "imported" by the countries of the EU. On the other hand, with the beginning of the crisis, Italian companies were confronted with a difficult choice: considerably reduce production or move production elsewhere to reduce production costs (Mariotti 2009). In fact, right after the crisis there was an increase in the weight of Italian FDIs in the East European Countries compared to the rest of the world, rising from 1.56% in 2006 to 6.42% in 2008 and 12.2% in 2010.

According to data published by Serbia Investment and Export Agency (SIEPA 2020) in 2015, in only the last 10 years Italian companies have invested over 3 billion euros in this country, opening about 600 companies and creating over 20,000 jobs, and have a turnover over 2.5 billion euros. Referring to the same source, it emerges that Italy has become the second most important investor after Austria.

Italian investments have also had the same trend in Albania. More than 440 Italian companies were opened in Albania and about 21,000 jobs were created by Italian companies in Albania in April 2017 in call centers alone, without overlooking those in the manufacturing sector (clothing industry) that employs over 120,000 people, a good part of which fulfill orders coming from Italy (INSTAT Albania 2021). Italy is the first investor and the first trading partner for Albania.

In Romania, Italy is the leading investor by numbers of registered companies (in 2015, 41,749 Italian companies were registered at the Trade Register, 20.41% of the total number of foreign companies registered in Romania), while it holds the eighth place among foreign investors by invested capital, over 3300 million euros in investments (Confindustria Balcani 2021). Referring to the same source, the Italian presence is also important in Bulgaria, with about 1000 Italian companies opened in this country, while it is less important in Bosnia and Herzegovina with 70 companies opened and only 2.8% of the foreign investments.

Regarding Italian companies that have moved to the Eastern European Countries, as is shown also by Cutrini and Spigarelli (2012), the sector that has shown greater delocalization dynamism is that of manufacturing (40%).

The foreign capital in the banking sector reaches up to 90% in countries such as Albania, Croatia, North Macedonia, and Bosnia and Herzegovina and, according to Bartlett and Prica (2013) and (Mariotti 2009), this capital is mostly Italian.

Figure 3 shows the trend of the delocalization of Italian companies for every 10,000 inhabitants in a given country for the period 2010–2016.

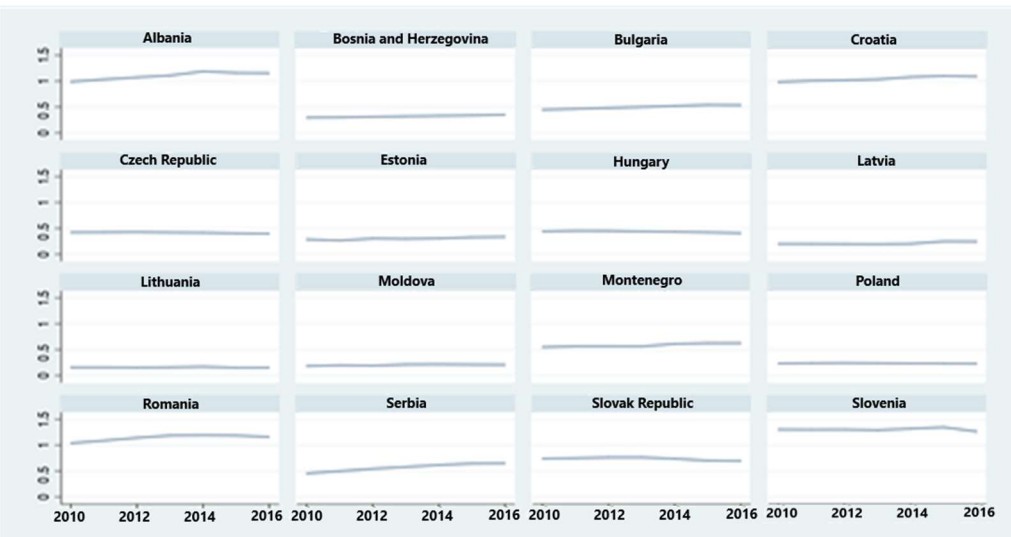

**Figure 3.** Density of Italian Firm Delocalization (out of 10,000 inhabitants). Source: ISTAT (2021).

The graph shows that the country with the highest presence of Italian companies for every 10,000 inhabitants is Slovenia, the EU country with the highest average salary in the Balkan area, 1756 euros. However, if we see data on turnover and employment of Italian companies in this country, Slovenia occupies the last place.

The number of Italian companies that have delocalized production to Slovenia is 270 with 5400 employees, with an average annual turnover of 1.3 billion euros.

Italian companies are more numerous in Romania, with about 2280 from 6534 companies in total that have delocalized to the East European Countries. The number of employees in Italian companies in Romania reaches about 96,031 workers out of 306,486 workers in total in all the Italian companies located in the East European Countries in 2017. The annual turnover of Italian companies in Romania reaches about 7.3 billion euros out of an annual turnover of 32 billion euros in the East European Countries. After Romania, the second most relevant country is Poland, with about 833 companies, 65,671 employers, and an annual turnover of 14.8 billion euros. The number of Italian companies in Poland is very low in comparison to Romania (1/3), whereas annual turnover is over 2/3 that of Romania, because the Italian companies operating in Poland are bigger compared to the Italian companies in Romania. Romania seems to have a very positive growing trend for the period 2010–2013 whereas, in the last two years, the trend is decreasing. On the other hand, this trend in Poland is constant.

The two East European Countries that attract the most Italian companies, in the period that we have considered, are Serbia and Albania. In Serbia, the average number of Italian companies that have delocalized production is 440, with over 19,000 employees and about 2.2 billion euros in annual turnover. In Albania, on the other hand, the number of Italian companies is around 350, the number of employees is about a quarter of that of Serbia, namely 5400, whereas the annual turnover is about one fifth of that of Serbia, 0.44 billion euros.

Figure 3 shows clearly that Croatia is another EU country liked by Italian companies. The number of Italian companies that have delocalized part of their production to this country is over 460. The number of employees is around 13,500 and annual turnover is over 1.3 billion. Considering ICE data, it is clear that the cost of labor is not the main factor that encourages Italian companies to delocalize to a particular country. If this were true, then the greatest concentration of Italian companies would be in Albania, as the country with the lowest level of average salary (419 euros).

Figure 4 presents other factors that may influence Italian companies to delocalize to one of the East European Countries.

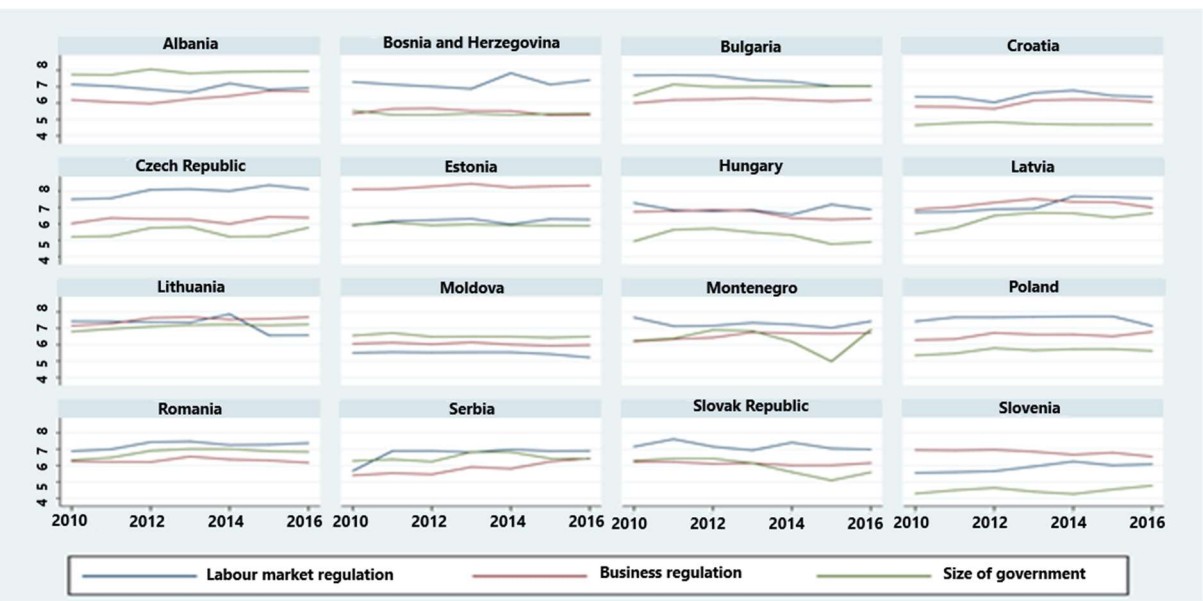

**Figure 4.** Factors influencing Italian firms' delocalization. Source: Economic Freedom of the World (2021).

Other factors that may influence Italian companies to delocalize to one of the East European Countries are "labor market regulation", "business regulation", and "size of government" (for all these indicators: 0 (too-much regulation) to 8 (few regulations, good for firms)).

The first involves: hiring and firing regulation, minimum wage, centralized collective bargaining, hours regulations, cost of worker dismissal, conscription.

The second involves: administrative requirements, bureaucracy costs, starting a business cost, extra payments/bribes/favoritism, licensing restrictions, tax compliance.

The third involves: government consumption, transfers and subsidies, government enterprises, top marginal tax rate.

In Figure 4 we see the different trends of labor market regulation, business regulation, and size of government by country. Focusing on the value of labor market regulation, it can be shown that the low labor market regulation in different East European Countries is associated with more Italian outsourced companies.

What is interesting about Figure 4 is the fact that the EU member countries have a high level of "labor market regulation". The difference between EU member countries and developing countries in Eastern Europe is low for this indicator.

Regarding "business regulations", the Baltic countries have high business regulations compared to the rest of the countries that we take into consideration in this study.

The correlation between "business regulation" and Italian enterprise delocalization in this country is the same as for the "labor market regulation".

Whereas, analyzing the indicator "size of government", we note that the lower the size of government, the higher the number of Italian companies that have delocalized. This is obviously an expected trend as a "smaller" government implies fewer taxes.

The Worldwide Governance Indicators are a research dataset provided by the World Bank Database (2021): standard data are scaled from −2.5 (bad) to 2.5 (good). It summarizes the views on the quality of governance provided by a large number of enterprise, citizen, and expert survey respondents in industrial and developing countries. These data are gathered from a number of survey institutes, think tanks, non-governmental organizations, international organizations, and private sector firms.

Figure 5 represents the estimation of institutional factors such as "political stability", "control of corruption", and "rule of law". The values for these three indicators vary from

high to low, that is, the lower the value, the less the perceived corruption, the better the political stability and the rule of law.

**Figure 5.** The Worldwide Governance Indicators. Source: The Worldwide Governance Indicators, World Bank Database (2018).

From the three last graphs we can say that, the sooner a country enters the EU, the better its institutional factors are. Albania and Macedonia are the countries with the highest corruption index and lowest political stability. That is why these are the last two countries from those taken into consideration that have not yet opened negotiations for accession to the EU.

To conclude, we can say that the delocalization of Italian companies has influenced the intensification of trade between Italy and East European Countries. Moreover, it has strengthened the links between companies operating in the various countries of the area, providing them with support and know how to deal with the neighboring markets.

## 4. Materials and Methods

The data used in this analysis allow us to estimate the factors influencing the delocalization of Italian companies to the East European Countries. They also give us some information about the reasons that make a country more attractive for Italian companies.

The time series for the countries taken into consideration is from 2010 to 2016. The number of observations in the complete panel, however, reaches 112 (Table 1). We preferred to use only ICE data because ICE is an official Italian Agency and the data are reliable, rather than data published by various associations which are somehow unreliable or do not match in terms of periods.

**Table 1.** Descriptive Statistics.

| Variables | Obs | Average | Min | Max | Dev. St |
|---|---|---|---|---|---|
| Delocalization density | 112 | 0.58 | 0.1 | 1.4 | 0.36 |
| Labour market regulation | 112 | 6.92 | 5.06 | 8.36 | 0.7 |
| Business regulation | 112 | 6.89 | 5.15 | 8.16 | 0.8 |
| Size of Government | 112 | 6.6 | 4.93 | 8.16 | 0.81 |
| Rule of Law | 112 | 0.37 | −0.52 | 1.37 | 0.55 |
| Control of corruption | 112 | 0.1 | −0.96 | 1.3 | 0.53 |
| Political Stability | 112 | 0.46 | −0.82 | 1.12 | 0.46 |
| Infrastructure (broadband) | 112 | 19.87 | 3.62 | 31.5 | 6.29 |



To carry out this study we used a balanced panel data of 16 countries from the East European Countries (Albania, Bulgaria, Bosnia and Herzegovina, Croatia, Czech Republic, Estonia, Hungary, Latvia, Lithuania, Moldova, Montenegro, Poland, Romania, Serbia, Slovak Republic, and Slovenia). We have excluded Turkey and Greece because are very different from the historical and economic point of view compared to the other East European Countries. We have also excluded Kosovo because it is a country that has proclaimed its independence relatively recently and, consequently, there are few data available and also the empirical tests for this did not make much sense.

Moreover, we excluded Armenia, Georgia, North Macedonia, and Ukraine because they show a very low level of Italian firms in contrast to the other countries included in the sample.

We set out to examine both the direct costs and the indirect costs that a company should sustain to delocalizes production in a given country using indicators such as: hiring and dismissal regulation, minimum wage, centralized collective contract negotiation, working hours regulations, cost of worker dismissal, conscription, administrative requirements, bureaucracy costs, starting a business cost, extra payments/bribes/favoritism, licensing restrictions, tax compliance, political stability, corruption, and broadband infrastructure.

The analysis will be conducted through a panel approach in which the dependent variable is the number of Italian companies in relation to the population.

The results of a standard Hausman test indicate the presence of subject-specific fixed effects. After the appropriate tests, we also found heteroscedasticity and autocorrelation in the residuals. Moreover, we tested for cross-sectional dependence by the CD-test described in Pesaran (2004, 2015). The results indicate that there is some (strong) correlation between panel units (cross-sectional dependence).

After transforming the data into a natural logarithm function to make them stationary, the econometric model is as follows:

$$LogDel_{it} = \beta_0 + \beta_1 logLab_{it} + \beta_2 logBus_{it} + \beta_3 logSize_{it} + \beta_4 logRule_{it} + \beta_5 logCorr_{it} + \beta_6 logPol_{it} + \beta_7 logInf_{it} + u_{it} \tag{1}$$

where Del—delocalization; Lab—Labor Market Regulation; Size—Government Size; Rule—Rule of Law; Corr—Control of Corruption; Pol—Political Stability; Inf—Infrastructure (broadband); $\beta_1$ to $\beta_7$ indicate the elasticity of Del with respect to the explanatory variables; $u_{it}$ is the error term with classical assumptions; country and time are shown by $i$ and $t$, respectively.

Hence, we perform a FE (within) regression with Driscoll and Kraay (1998) standard errors (Hoechle 2007). The error structure is assumed to be heteroscedastic, autocorrelated, and correlated between the groups (panels). Finally, we use the small sample adjustment, accounting for the small size of our sample.

In order to operate with stationary variables and in order to allow a clear quantitative interpretation of the results, we use the natural logarithms of all the variables. In the panel analysis we include time and country fixed effects; when both the dependent variable and the regressors are in log, if we change X by one percent, we expect Y to change by β percent, where β is the estimated coefficient. Such relationships, where both Y and X are log-transformed, are commonly referred to as elastic in econometrics, and the coefficient of log X is referred to as the elasticity.

In addition to the previous variables, among the regressors we also consider a variable accounting for the quality of infrastructure in the country: "Fixed broadband subscriptions (per 100 people)" (World Bank data).

Our main goal is to empirically verify the impact of labor market regulation, business regulation, and institutional factors, on the delocalization of Italian companies on East European Countries. Specifically, the impact of labor costs, which have been cited (Helg and Tajoli 2005; Cietta 2008; De Nardis and Traù 2005; Smith and Pickles 2011) as the main factor which explains the international division of production.

We want to verify what relevant authors have said about the impact of "labor market regulation" in delocalization of enterprises (Baylos 2013; Cardullo et al. 2013).

Finally, special attention will also be given to the institutional factors that influence Italian companies to delocalize to one country rather than another, namely the impact of "corruption", the "rule of law", and the "political stability", and finally the "broadband infrastructure" of the host country. This empirical analysis aims to understand if Italian companies take into consideration only the countries with the lowest direct costs or prefer the East European Countries that are already part of the EU and have less corruption and more political stability.

The number of Italian companies (we have considered only Italian companies that are branches of Italian companies with revenue of not lower than 500,000 euros) which have delocalized their production for every 10,000 inhabitants in the seventeen East European Countries that we have taken into consideration serves as dependent variable, whereas economic and institutional factors serve as our independent variables.

The positive aspect about this panel data is the homogeneity of the data. The negative aspect is that we cannot extend the panel data to include previous years, as such data are not available.

It would be very interesting to verify empirically whether the delocalization of Italian production would create added value for the Italian economy in general or not. However, this verification is impossible because we can never know for sure if the jobs opened in the East European Countries by Italian companies have been accompanied by a certain number of job losses in Italy. In addition, there are no data available on the number of unemployed and the unemployment period of Italian workers who have lost their jobs due to the delocalization of the Italian companies to the East European Countries.

We did not take into consideration the distance between Italy and each individual country because we thought that the distances are very similar and would not change transport costs much.

It would also be interesting to measure the change in the productivity of the host country once the Italian company has moved production there. However, the host countries did not have relevant data on productivity by sector for the period in question.

## 5. Results

The empirical results of the impact of "labor market regulation", "business regulation", "size of government", "rule of law", "control of corruption", "political stability", and "broadband infrastructure" in the delocalization of Italian companies to the East European Countries are show in Table 2.

**Table 2.** Panel Analysis results.

| Dep Var: Italian Firms Out of 10,000 Inhabitants | Panel Regression Driscoll–Kraay Standard Errors | Panel Regression Driscoll–Kraay Standard Errors |
|---|---|---|
| Labor Market Regulation | 0.30 ** (0.101) | 0.32 ** (0.089) |
| Business Regulation | 0.39 ** (0.136) | 0.38 ** (0.134) |
| Size of Government | 0.13 * (0.058) | 0.13 ** (0.055) |
| Rule of Law | 0.38 *** (0.078) | 0.41 ** (0.117) |
| Control of Corruption | −0.28 ** (0.105) | −0.27 ** (0.106) |
| Political Stability | 0.05 (0.068) | |
| "Infrastructure" (broadband) | 0.17 *** (0.037) | 0.19 *** (0.029) |
| Constant | −2.98 *** (0.352) | −3.04 *** (0.349) |
| Country Fixed Effects | Yes | yes |
| Year Fixed Effects | Yes | yes |
| F(12,6); Prob > F | 71.14; 0.000 | 133.86; 0.000 |
| within $R^2$ | 0.26 | 0.13 |
| Countries | 16 | 16 |
| Years | 7 | 7 |
| Observations | 112 | 112 |

Robust standard error in brackets; sign: *: 10%; **: 5%; ***: 1%.

The panel analysis indicates that the "labor market regulation", "business regulation", and "size of government" are correlated with the presence of Italian firms for 10,000 inhabitants. These variables have a positive and significant effect.

From the table of the panel analysis results it can be shown that lower levels of "labor market regulation", "business regulation", "size of government" correspond to a higher number of Italian companies present in each country. This means businesses need a labor market through which labor costs can be reduced and where the system's legal regulation gives businesses the opportunity to maneuver, especially in cases of legal responsibilities. On the other hand, businesses need an elastic, non-bureaucratic government from which quick and clear decisions are expected without high costs.

This result confirms the expectations mentioned in the literature cited in the previous paragraphs: the low labor costs in a certain country encourages Italian companies to delocalize production to that country.

The panel analysis indicates also that the "rule of law" has a positive and significant effect, whereas the "control of corruption" has a negative significant effect.

This result indicates that the Italian companies are looking for an environment in which the state can enforce its own rules, but also a state that is "corruptible" (so the rules bend to the will of the companies).

To conclude, we can say that the explanatory variable, political stability, has an insignificant correlation with the dependent variable.

Finally, the independent variable "broadband Infrastructure" has an important significant correlation with enterprise delocalization: the better the infrastructures in the East European Countries, the more Italian companies delocalize to that country. Of course, such a result is expected, since the infrastructure helps transport development, the communication increases, and therefore the business development costs are reduced.

## 6. Discussion

In this paper we examined the process of delocalization of Italian companies and empirically verified the main factors that affect the delocalization of Italian companies to a particular country in the East European Countries.

Thus, this empirical analysis confirms what has been said by most of the economic literature in this field, i.e., that labor costs are one of the main factors which explain the international division of production.

From the analysis of the empirical literature on the factors influencing the delocalization, and from the descriptive analysis of official data obtained by the ICE, we can conclude that Italian companies tend to replace part of their unskilled labor force in Italy with that of the East European Countries; however, we confirm that the delocalization of Italian companies in the East European Countries has no significant implications for the job losses in Italy.

Our research contributes to the debate on the factors that influence the delocalization of Italian companies.

It contributes to two main directions. First, it is a case study, at a macroeconomic level, of the factors influencing the delocalization of production, taking into consideration the countries that are already part of the EU as well as those who want to join it. This serves to explain the economic and institutional differences that exist in different countries with different levels of integration (Zhelev and Tzanov 2012; Landesmann and Wörz 2006).

Finally, this paper demonstrates the interesting fact that the delocalization of production is not negatively affected by the corruption of the host country. This means that sometimes the companies that delocalize production do not mind a bit of corruption (Labrianidis 2001).

## 7. Summary

By analyzing the delocalization process of Italian companies and empirically verifying the main factors that affect the delocalization of Italian companies to a particular country

in the East European Countries, we have examined the factors that influence this process through two different groups of factors.

The first group examines the impact of economic factors such as "labor market regulation", "business regulation", and "size of government", of the host country, that encourage Italian enterprises to delocalize production.

The second group examines the impact of institutional factors such as "rule of law", "control of corruption" "political stability", and "broadband infrastructure" on the delocalization of Italian companies to the East European Countries.

The empirical analysis has shown that the low labor cost of a certain country encourages Italian companies to delocalize production to that country.

Moreover, the empirical results show that even institutional factors such as "rule of law" have a positive and significant effect, whereas the "control of corruption" has a negative significant effect. The result of the correlation of the two variables mentioned above indicates that the Italian companies are looking for an environment in which the state can enforce its own rules, but also a state that is somewhat "corruptible" (so the rules "bend" according to the will of the companies).

Finally, this paper also examined the impact of "broadband infrastructure" on the delocalization of Italian companies in East European Countries. The results in this case suggest that, the better the infrastructures in an East European Country, the more Italian companies delocalize in that country.

To conclude, we can say that the explanatory variable, "political stability", has an insignificant correlation with the dependent variable.

A future study would be the analysis of the balance of costs and benefits that the delocalization of production by Italian companies causes in Italy. While studying and examining microeconomic indicators such as: the costs of dismissal of domestic workers, pay cuts resulting from job losses in Italy, and the higher revenue for the Italian state deriving from the higher profits of Italian companies resulting from lower production costs abroad, we would understand if Italy stood to win or lose from delocalization.

**Author Contributions:** Conceptualization, K.M. and E.K.; methodology, K.M.; software, E.K.; validation, K.M. and E.K.; formal analysis, E.K.; investigation, K.M.; resources, K.M. and E.K.; data curation, E.K.; writing—original draft preparation, E.K.; writing—review and editing, K.M.; visualization, K.M.; supervision, K.M.; project administration, E.K.; funding acquisition, K.M. All authors have read and agreed to the published version of the manuscript.

**Funding:** The publication of this study was fully financed by the Catholic University Our Lady of Good Counsel.

**Informed Consent Statement:** Not applicable.

**Data Availability Statement:** Not applicable.

**Conflicts of Interest:** The authors declare no conflict of interest.

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
