# Peer review of "The Determining Factors of the Delocalization of Italian Companies to East European Countries"

_economies, doi:10.3390/economies11110273_

Round 1

Reviewer 1 Report

The paper provides interesting findings. To improve the paper, I suggest the following:

Technical details:

- Improve writing style: Avoid paragraphs that contain only one sentence (for example, first paragraph in the Introduction). Rather connect paragraphs that address similar issues/topics and try to use message-driven technique.

- Improve language - for example, this sentence is not clear: The methodology used in this study provides firstly a review of empirical studies on 39 delocalization, and not only. The paper should be proofread.

Content of the paper:

Introduction:

- Explain in more detail why studying the delocalization of Italian companies to East European countries is important (currently no relevant data). How big is this phenomenon in scale, any other peculiarities?

- Improve description of methodology - no details about the methods used.

- Chapter 1.1.: I suggest to present this chapter as a separate chapter 2 titled 2 Literature Review

- Chapter 1.2.: 1) This should be a separate chapter and not part of the Introduction. 2) I suggest reorganizing the chapter in a way that statistics are presented graphically, comparing all countries included in the analysis. In this way, the reader will get a better understanding of the current situation and movements of chosen indicators. 3) In the introduction of the chapter, state according to which indicators are you analyzing the co-operation between countries and why these indicators are relevant. 4) Point on the distinction between EE that are members of the EU and others. In my opinion, this is very relevant and influences the delocalization choices of companies. 

Materials and Methods: 

There are no details about the method applied. Describe the panel regression framework used, by formally presenting and explaining also the regression equations/models used. The authors describe this in Results, yet it should be in this chapter.

All methods applied in section Results should be described in this section. 

Results:

- There is no need for repetition of details regarding samples, data sources... Focus on the presentation of results.

- Figures: Improve the titles (the title should describe the content of the figure); figures should be larger.

- Figures 2 and 3: Improve the content interpretation of the results and dynamics of certain indicators in certain countries. In this way the figure gest value.

- Panel regression analysis: 1) details about the methodological framework should be moved to the section above; 2) there is no need to give an explanation regarding the interpretation of the regressors (the paper is aimed at the academic public!); 3) in section Methods present the formal regression equation and add explanation regarding the variables (how they are measures, also why they are relevant); 4) I believe there should be a dummy variable controlling also for membership in the EU; re-consider also other relevant variables that will control for differences between countries (for example, GDP, labor market situation, currency, distance...)

- Results should be interpreted in more detail, not only their sign. 

Discussion:

Add how the findings of the paper compare and contribute to the existing literature (with adding also references to the existing papers). 

Point on the value added of the paper.

The text should be proofread.

Reviewer 2 Report

Review of the paper titled „The Determining Factors of the Delocalization of Italian Companies to East European” under consideration to Economies journal:

Major issues:

-         It is advisable to introduce the importance of the matter slightly better the research is about. In your case please add 1-2 paragraphs about the significance of the firm delocalization, GVS, fragmentation of work, etc.

-         Please try to reformulate the goal, for instance: Paper aims at identification of factors affecting …. or The main goal of the paper is to identify the extent to which economic, institutional, …, factors drive delocalization of Italian firms to EEC.

-         In the introduction, please follow the scheme: (i) significance of the matter, (ii) state of the art (missing in the paper), (iii) your approach, (iv) data and methods, (v) resume of findings and implications,  (vi) agenda. At this point, you should consider writing a part on the current state of the art, then adding what’s new in your paper, later reshaping methodology and your approach into a more consistent matter.

-         In introduction write about the research sample, its size, and destination countries.  

-         In the data and methods section, one should write about the econometric approach utilized, why is sufficient or superior (compared to other approaches) to use in that study. I cannot see descriptive statistics of the data used in this study, as well. Please add these information.

-         The charts 1-3 are not visible well, due to the scale and their sizes. Please try to present the data in a different manner or change the scale. Maybe prepare tables with appropriate data, like level, share, and changes in years X-Y. This section is not results, but stylized facts on Italian firms’ delocalization. Consider, making this section more condense, as it is lengthy and presenting only additional information.

-         Line 477: You have a panel dataset. Please write is a balanced or unbalanced panel. The following lines up to 500 should be in the data and methods section. Please write why this approach fits your dataset, the matter you want to verify, and why it is better than other approaches. In this section, you should also provide the table with descriptive statistics of the variables used in the study, their origin, coding, etc. Thanks to that your paper will be shorter.

-         The paper lacks a robustness test or sensitivity analysis. It would be advisable if your result would hold if you have used another econometric approach or set of variables, etc. Since you operate on data that endogeneity may cause some concerns, you may use e.g. system GMM approach (i.e. two-stage). It’s a dynamic approach in which you add a lagged dep var into the descriptives. In STATA, you may call it by xtabond2 routine.

-         The results should be interpreted in a more detailed way. Presenting possible factors or aftermaths – in short, why you think it is so. At this point this section lacks economic interpretation.

-         There is no summary at the end of the paper. The existing discussion is more like a summary.

-         There is no discussion section. In such section the author compares his/her/their results with other empirical evidence, done so far; and states if their results are in line with or against the existing state of the art.

-         In summary please write about policy recommendations stemming from your results.

Minor issues:

-         Please try to build longer paragraphs, like at least 4-5 lines of text; in the introduction for instance.

-         It’s a pity that you don’t have firm-level data. Then, the identification of particular inner or outer factors would be more profound. I understand that the availability of such data is somewhat difficult. Another approach would be to use the gravity model to estimate factors affecting the number of firms delocalizing in each year to particular countries from your sample.

-         Considering the above, then the square of the distance between Italy and other countries, could be used as one of the descriptives, similar to other distances (economic, legal, institutional, etc.).

-         Below we see -> don’t use such statements. Instead, refer to a particular chart or table.

Outcome:

My opinion about the paper is rather positive. It undertakes an interesting and important matter. However, there is still a lot of work to be done in order to increase the paper credibility. First of all, the structure of the paper is wrong and it needs refinements in order to make it more clear for the reader. Some shifts between parts of the manuscript are needed. Secondly, new stylized facts section should be introduced and shortened from the original manuscript. One should also work on the text formatting, which seems to not fit the template well (frequent short paragraphs). Finally, it is advisable to add some robustness test, discussion of the results, and extend the results and implications stemming from the obtained estimates.

Round 2

Reviewer 1 Report

None